

# Insights into the morphology of multicomponent organic/inorganic aerosols from molecular dynamics simulations

Katerina S. Karadima,[1,2] Vlasis G. Mavrantzas,[1,2,3] Spyros N. Pandis[1,2,4]

[1]Department of Chemical Engineering, University of Patras, Patras, GR 26504, Greece

[2]Institute of Chemical Engineering Sciences (ICE–HT/FORTH), Patras, GR 26504, Greece

[3]Department of Mechanical and Process Engineering, ETH Zürich, CH 8092 Zürich, Switzerland

[4]Department of Chemical Engineering, Carnegie Mellon University, Pittsburgh, PA 15213, USA

*Correspondence to*: Spyros N. Pandis (spyros@chemeng.upatras.gr)

**Abstract.** We explore the morphologies of multicomponent nanoparticles through atomistic molecular dynamics simulations under atmospherically relevant conditions. The particles investigated consist of both organic (cis-pinonic acid/CPA, 3-methyl-1,2,3-butanetricarboxylic acid/MBTCA, $n$-$C_{20}H_{42}$, $n$-$C_{24}H_{50}$, $n$-$C_{30}H_{62}$ or mixtures thereof) and inorganic (sulfate, ammonium and water) compounds. The effect of relative humidity, organic mass content and type of organic compound present in the nanoparticle is investigated. Phase separation is predicted for almost all simulated nanoparticles either between organics and inorganics or between hydrophobic and hydrophilic constituents. For oxygenated organics, our simulations predict an enrichment of the nanoparticle surface in organics, often in the form of islands depending on the level of humidity and organic mass fraction, giving rise to core–shell structures. In several cases the organics separate from the inorganics, especially from the ions. For particles containing water–insoluble linear alkanes, separate hydrophobic and hydrophilic domains are predicted to develop. The surface partitioning of organics is enhanced as the humidity increases. The presence of organics in the interior of the nanoparticle increases as their overall mass fraction in the nanoparticle increases, but this depends also on the humidity conditions. Apart from the organics–inorganics and hydrophobics–hydrophilics separation, our



simulations predict a third type of separation (layering) between CPA and MBTCA molecules under certain conditions.

## 1 Introduction

Atmospheric aerosol particles exert a major impact on human health (World Health Organization, 2016), air quality (Fuzzi et al., 2015), and visibility (Wang et al., 2015). They also impact Earth's radiation balance and consequently global climate, but their contribution to climate forcing remains highly uncertain (IPCC, 2013).

Atmospheric particles typically contain both organic and inorganic compounds. The fine inorganic mass fraction is mainly inorganic salts, with ammonium sulfate being the most abundant (Zhang et al., 2007). The contribution of organics to sub–micrometre atmospheric particulate matter mass can be as high as 90%. The thousands of atmospherically–relevant organic compounds differ significantly in volatility, solubility and other physicochemical properties, and their complexity hinders our overall understanding of atmospheric organic aerosol (Glasius and Goldstein, 2016).

During its lifetime, atmospheric particulate matter evaporates or grows by condensation but also undergoes chemical transformations due to a number of heterogeneous reactions (Buseck and Adachi, 2008; Seinfeld and Pandis, 2006). Aerosol phase and morphology, among others, can play a significant role in these processes. Specific morphologies, such as coating by organic films, may affect water uptake (Davies et al., 2013), the rate of heterogeneous reactions (Folkers et al., 2003; Zhou et al., 2012) or the formation of cloud droplets (Sareen et al., 2013). Aerosol microstructure can also have a strong effect on bulk and interfacial diffusion rates (Shiraiwa et al., 2011) or on processes such as light scattering and absorption (Takahama et al., 2010; Zhang and Thompson, 2014).

Several types of morphologies and different phase states have been observed in ambient particles or model systems. Virtanen et al. (2010) reported that biogenic secondary organic aerosol (SOA) particles often exist in an amorphous solid state. On the other hand, the majority of the highly oxygenated or secondary organic particles investigated by Bahadur et al. (2010) and Takahama et al. (2010) were liquid–like and



almost spherical, and carboxylic surface enrichment was observed. Enhanced surface partitioning has also been reported by Werner et al. (2016) for succinic acid. The hydrophobic $D_{62}$-squalene and $\alpha$-pinene oxidation products form separate phases in the form of a core–shell morphology according to Robinson et al. (2015). Veghte et

al. (2014) have observed homogeneous morphologies for highly water–soluble dicarboxylic acids and partially engulfed ones for acids with limited solubility. Partially engulfed phases have also been reported by Dennis-Smither et al. (2012) in deliquesced oleic acid particles. Reid et al. (2011) stressed the importance of morphologies that include hydrophilic and hydrophobic domains of partially engulfed

structures.

Aerosol particle morphology and phase state are often reported to be humidity–dependent (Dennis-Smither et al., 2012; Werner et al., 2016). In–between the several humidity cycles (efflorescence and deliquescence) that the aerosol particles undergo, liquid–liquid phase separation may occur (Bertram et al., 2011;

O'Brien et al., 2015; You et al., 2012) with one liquid phase rich in organic and the other rich in inorganic (water and inorganic salts). The liquid–liquid phase separation of organic aerosols is usually explained as a salting–out effect: the decreased solubility of organics in water due to inorganic salts present forces them to accumulate together into a separate phase (Bertram et al., 2011; You et al., 2014).

Among several factors (such as O:C ratio, functional groups in the organic compounds, organic–to–inorganic mass ratio, type of inorganic salts present, temperature and particle size) that influence liquid–liquid phase separation, the O:C appears to be the most important (Song et al., 2012; 2013; You et al., 2014; You and Bertram, 2015). Liquid–liquid phase separation has always been reported for organic

compounds O:C lower than 0.5 while it has never been observed for values of the O:C higher than 0.8. For intermediate values of the O:C, liquid–liquid phase separation depends on the type of functional groups present.

Thermodynamic models can predict the morphology of mixed organic/inorganic aerosol particles based on calculations of the total surface free

energy of the different phases (Kwamena et al., 2010; Reid et al., 2011; Qiu and Molinero, 2015). Models that describe the formation of surface films by fatty acids on




the aerosol surface (Seidl, 2000) have also been proposed. Complex multicomponent aerosol systems are typically described by a combination of theoretical and kinetic models (Shiraiwa et al., 2013). This last study highlighted the importance of phase state and particle morphology or microstructure on gas–particle partitioning; it has

also questioned a number of common assumptions typically invoked by some aerosol models, such as the ideal mixing approximation of organics (namely that organic compounds form ideal mixtures with other organics, inorganic salts and water) and the assumption of an instantaneous equilibrium established between gas and particle phases. For example, some air quality models assume a single phase composed of

organics, inorganics and water (Pun et al., 2003). In others, two separated phases, an organic and an aqueous, are assumed (Koo et al., 2003). Clearly, these assumptions affect the model predictions for the partitioning of semi-volatile compounds between gas and particulate phases and the particle properties.

By addressing the detailed molecular structure and motion of nanoparticles at

the level of individual atoms, molecular dynamics (MD) can provide unique information on molecular organization, morphology and microscopic dynamics of ultrafine aerosol. Previous applications of the MD method (Chakraborty and Zachariah, 2011; Hede et al., 2011; Li et al., 2010; Ma et al., 2011) have examined slabs or particles composed only of organic acids and water. More recently, Liyana-

Arachchi et al. (2014) explored salt water slab interfaces with normal alkanes and sodium dodecyl sulfate, while Hudait and Molinero (2014) studied the internal structure of ice–crystallized ultrafine water–salt aerosols. MD was also used by Qiu and Molinero (2015) to verify predictions regarding the liquid–liquid phase separation of nonane and water droplets. In the last two studies, a united–atom representation

was invoked.

The types of particle morphologies observed so far include partial engulfing, core–shell morphology with an inorganic core and an organic coating, and several intermediate (mixed) formations, such as surfaces enriched in organics, morphologies with additional inorganic or organic (liquid either glassy) inclusions, and

morphologies characterized by concentration gradients. However, the conditions under which these morphologies appear are not as yet clear. In the present work, we



investigate the morphology and microstructure of aerosol nanoparticles through MD simulations under atmospherically relevant conditions. We focus on particles composed of organic–inorganic mixtures under prespecified conditions of relative humidity corresponding to freshly formed or aged, primary or secondary, aerosol. Our

emphasis is on the local microscopic structure and overall molecular organization (morphology) of the formed nanoparticles as well as on the key microscopic interactions that lead to these structures. We focus on the effects of relative humidity, type of organic compound present and organic mass fraction on nanoparticle structure.

**2 Particle composition, simulation conditions and methods**

**2.1 Particle components**

Five organic compounds with different physicochemical properties were selected and studied (either separately or in mixtures) as representatives of the organic content of atmospheric particles. They include cis-pinonic acid (CPA) or $C_{10}H_{16}O_3$, 3-methyl-

15 1,2,3-butanetricarboxylic acid (MBTCA) or $C_8H_{12}O_6$ and the three saturated alkanes $n$-eicosane or $n$-$C_{20}H_{42}$, $n$-tetracosane or $n$-$C_{24}H_{50}$, and $n$-triacontane or $n$-$C_{30}H_{62}$. CPA and MBTCA are oxidation products of monoterpenes (Müller et al., 2012; Zhang et al., 2015) and are used as tracers of biogenic secondary organic aerosol in field measurements (Cheng et al., 2011; Szmigielski et al., 2007). MBTCA is very

soluble in water (solubility about 320 g $L^{-1}$) (Dette et al., 2014) due to the presence of the three carboxylic groups, while CPA dissolves less (solubility around 3.7 g $L^{-1}$) (Hyvärinen et al., 2006). The O:C of CPA and MBTCA is 0.3 and 0.75, respectively. On the other hand, the three saturated alkanes ($n$-$C_{20}H_{42}$, $n$-$C_{24}H_{50}$, $n$-$C_{30}H_{62}$) which were selected for their hydrophobic character are components of anthropogenic but

also biogenic primary organic aerosol (Kotianová et al., 2008; Wang et al., 2009). Normal alkanes are water insoluble organic components (with O:C = 0) and form macroscopic waxy crystals (Qiu and Molinero, 2017; Mavrantza et al., 2001). Water, sulfate and ammonium ions were chosen as the typical inorganic components in our study, while the surrounding atmosphere was represented by nitrogen ($N_2$) and

oxygen ($O_2$) at their atmospherically relevant stoichiometry.





In all cases, 200 sulfate anions and 400 ammonium cations were placed in the simulation cell. Depending on the simulation, 10–400 organic molecules and 400–3600 water molecules were also considered. The total number of $N_2$ and $O_2$ molecules, which controls the size of the simulation cell in an $NpT$ simulation
(Karadima et al., 2017), varied from 300 to 900. The composition of the simulated particles, their corresponding wet and dry organic mass fractions and relative humidity conditions are summarized in Table 1.

### 2.2 Simulation conditions and protocol

The Materials and Processes Simulations Platform (MAPS®) (Scienomics SARL,
2015) was used to construct all molecules and ions, and to build initial configurations for the systems of interest. The simulations were executed using cubic simulation cells subject to periodic boundary conditions in all three directions with an initial edge length of 150 or 170 Å depending on the total number of molecules/ions considered. Prior to all MD simulations, the initial configuration was subjected to potential energy
minimization in order to obtain a structure that is completely free of atom overlaps. The MD simulation was performed in the isothermal–isobaric ($NpT$) statistical ensemble (at pressure $p = 1$ atm and temperature $T = 320$ K). The relatively high temperature was chosen to ensure ergodicity of the MD simulations (Karadima et al., 2017), as dynamics can become prohibitively slow at lower temperatures. The effect
of this choice will be addressed in selected sensitivity simulations. The Nosé–Hoover thermostat–barostat (Hoover, 1985; Nosé, 1984) and the velocity–Verlet integrator (Verlet, 1967; 1968) were employed, with the integration time step for the dynamic equations of motion set equal to 1 fs. All simulations were performed with the open–source software Large–scale Atomic/Molecular Massively Parallel Simulator
(LAMMPS) (Plimpton, 1995). The total simulation time was in all cases 60 ns, of which only the last 10 ns were used for analysis with atomic positions stored every 10 ps. For the visualization of the simulation trajectories, the open–source software BIOVIA Discovery Studio® Visualizer (Dassault Systèmes BIOVIA Discovery Visualizer Studios, 2016) was used.



The all–atom OPLS (Optimized Potentials for Liquid Simulations) force field (Jorgensen and Tirado-Rives, 1988) was adopted, except for sulfate ions, where the OPLS parameters were modified as described by Karadima et al. (2017). All parameters are available at the Supplementary Information of Karadima et al. (2017).

Additional parameters for MBTCA can be found in the Section S1 of the Supplement of the current study. The SPC/E model (Berendsen et al., 1987) was used for water and a two–site model (Tsolou and Mavrantzas, 2008; Wang et al., 2013) was adopted for treating nitrogen and oxygen molecules. The bond lengths of water, nitrogen and oxygen molecules, and of the bond bending angles of water molecules were kept

constant in the simulations using the SHAKE algorithm (Miyamoto and Kollman, 1992). The particle–particle particle–mesh (PPPM) method (Eastwood et al., 1980) was employed for computing electrostatic interactions with calculations carried out in real space up to 14 Å, and in the reciprocal Fourier space beyond that. The van der Waals interactions were modelled with a 12–6 Lennard–Jones potential up to 12 Å

and a switching–shift function (Brooks et al., 1983) between 12 and 14 Å. Special interactions and Lennard–Jones pair coefficients between different types of atoms were calculated following the mixing rules as prescribed by the OPLS force field (Jorgensen and Tirado-Rives, 1988).

Aerosol nanoparticles were allowed to form by following the procedure

outlined in detail in Karadima et al. (2017). Briefly, the nanoparticle was allowed to form spontaneously in the course of the MD simulation, starting from an initial system configuration wherein molecules had been placed randomly in the simulation cell. Usually, a single nanoparticle formed after 1 to 10 ns of simulation time. Particle equilibration and relaxation were examined by monitoring the time evolution of

several quantities: (i) the radius–of–gyration of the particle, (ii) the mean square displacement of the centres–of–mass of two molecules/ions with respect to their separation at zero time in the particulate phase, and (iii) the time autocorrelation function of the angle formed by the position vectors of two tagged atoms in the organic molecules. Typically, the equilibration period took 20 to 30 ns after particle

formation. Thus, in our study, only the last 10 ns of the simulation were used for the



calculation of average values corresponding to a nanoparticle of a fully equilibrated microstructure.

## 2.3 Particle morphology

The equilibrium value of the relative humidity for each aerosol particle formed was
5 estimated from the knowledge of its composition and the average number of water molecules in the particle phase (Table 1) using the Extended Aerosol Inorganics Model (E-AIM) (Clegg et al., 1992; 1998; Wexler and Clegg, 2002).

In the course of the MD simulations, the shape of the formed nanoparticle often deviated from the spherical one, being closer to an ellipsoid. To quantify its
degree of sphericity (or, equivalently, asphericity), we computed its three major semi–axes from its mass inertia tensor (Goldstein et al., 2007). Then, to characterize the particle shape, we used the relative shape anisotropy factor $\kappa^2$ (Theodorou and Suter, 1985) (Table 1). A perfect sphere has a $\kappa^2$ equal to 0 while values equal to 1 correspond to linear symmetry (Theodorou and Suter, 1985). From the computed
values of the three major semi–axes, the volume and average density of the nanoparticle were estimated. From the latter, we calculated next the wet diameter of the corresponding equivalent spherical particle (Table 1).

For the purposes of this study, we considered that the particle volume can be divided in three domains: the particle core (the space around its centre–of–mass
covering 35% of its volume), the intermediate ring–like region of the particle (the space right after its core covering an additional 35% of its volume), and the surface of the particle (its outer–most region, with a volume equal to 30% of the total particle volume). The space division was accomplished by considering the ellipsoidal shape of the particle (Fig. S1) and drawing ellipsoids around its centre–of–mass with semi–
axes defined in proportion to those computed by the mass inertia tensor of the entire particle. According to such a volume division, the thickness of the surface region is limited to 3–10 Å, depending on the exact size of the particle. More details regarding these geometric calculations are available in Section S2 of the Supplement. The presence of the various molecules in the three domains was quantified by calculating
the corresponding percentages of their atoms in these regions. This calculation does



not take into account whether all the atoms of a specific molecule are located in the examined region, but instead considers only the number of atoms found there. The spatial distribution of atoms inside the particle was also examined by computing radial density profiles (i.e., profiles as a function of radial distance from the centre–

of–mass of the particle).

Spatial correlations between the particle components were quantified through calculations of pair correlation functions (or radial distributions functions), $g(r)$; these functions are appropriately normalized (Allen and Tildesley, 1991) and provide a measure of the probability of finding an atom at a certain distance (within a tolerance)

from a reference atom in the particle. The values of these functions at short distances provide also a measure of the degree or strength of local packing in the particle.

The values of all the above–mentioned properties were computed as ensemble averages over the atomistic configurations accumulated during the last 10 ns of the simulation.

## 15   3 Results and discussion

### 3.1 Effect of organic compound type

In the first series of simulations (simulations No 1–5 in Table 1), 10 molecules of CPA or MBTCA or one of the three normal alkanes were considered together with 200 sulfate anions, 400 ammonium cations and 400 water molecules. The formed

nanoparticles had a low (7–14 % dry) organic mass fraction. The relative humidity was 40 % (Table 1) and the equivalent particle diameters around 4.3 nm, corresponding to a density of 1.4 g cm$^{-3}$.

In all cases, more than 60 % of the organic molecules were found to reside at the surface of the particle (Fig. 1). *N*-triacontane was also found deep in the particle

with around 40 % of its mass located in the two inner domains; for the rest of the compounds, this percentage was lower, around 20 %. This difference is due to the molecular length of *n*-triacontane, which makes it comparable to the size of the nanoparticle.

Around 10 % of the MBTCA molecules were observed to reside at the core of

the particle and another 10 % in the intermediate ring–like domain. The percentage of



MBTCA and CPA molecules in these two innermost regions was basically the same, but more MBTCA could be found a little deeper than CPA. This is again a chain length effect; it is related to the smaller size of MBTCA compared to CPA molecules.

Most of the organic mass for all compound types examined was at the surface. Typical snapshots of these nanoparticles are shown in Fig. 2. The two oxygenated molecules, CPA and MBTCA, were either alone or in groups of 2–3 molecules forming small neighoubrhoods or islands at the surface of the particle (Fig. 2a–b). For the alkanes, organized structures were observed that were more and more robust as the number of carbon atoms increased. The local structures formed by *n*-eicosane were not stable and broke down during the simulation (Fig. 2c–d). *N*-tetracosane molecules showed a tendency to pack together, but they were also found at a certain distance from each other (Fig. 2e). *N*-triacontane molecules, on the other hand, formed a stable, highly–organized structure (Fig. 2f).

The morphologies observed for *n*-eicosane and *n*-triacontane are consistent with those reported for the NaCl water slabs studied by Liyana-Arachchi et al. (2014). The different alkane structures observed at low organic mass content can be related to their melting points. For *n*-eicosane, this is 310 K, i.e. lower than the simulation temperature; for *n*-tetracosane, it is close to the simulation temperature, and for *n*-triacontane it is above the simulation temperature (Smolenskii et al., 2002).

Inorganic species were observed in all regions with the core of the particle containing almost exclusively inorganic compounds (Fig. S2). In all three nanoparticle domains (and for all particles studied here), the ratio of ammonium–to–sulfate ions was practically equal to two (1.99 ± 0.05), as expected. More than 65% of the water molecules were in the surface region in all cases and less than 15% could be found in the particle core (Fig. S3).

For all simulated particles except the *n*-eicosane–based ones, all organic molecules in the course of the MD simulation remained in the particulate phase. For the *n*-eicosane simulation, two eicosane molecules were close to migrating from the condensed phase to the gas phase (Fig. S4a–b) and eventually one of them succeeded in jumping from the particulate phase to the gas where it remained for the rest of the simulation (Fig. S4c–d); the other molecule continued to be in the particle phase. We





note that *n*-eicosane had the highest vapour pressure of all the normal alkanes examined here.

## 3.2 Effect of relative humidity

The particles discussed in the previous section were at relatively low relative humidity

(RH) conditions. To investigate the effect of RH on particle morphology, additional simulations were performed under higher RH levels.

### 3.2.1 CPA–containing particles

We studied the morphology and structure of CPA–containing particles at increasing levels of RH, from 40 to 88 % (simulations No 6–9 in Table 1), by increasing the

10 number of water molecules in the simulation cell from 400 to 3200 while keeping constant the number of CPA molecules (10) and ions (200 sulfate and 400 ammonium ones). The equivalent particle diameter increased with increasing RH, reaching almost 6 nm for the largest particle. The mean particle density decreased from 1.5 g cm$^{-3}$ at 56 % RH to 1.3 g cm$^{-3}$ at 88 % RH. The nanoparticles formed at higher RH were

15 more spherical (the relative shape anisotropy factor $\kappa^2$ decreased almost to zero, see Table 1).

According to our simulations, as the RH increases, less CPA is found in the two innermost domains of the particle (Fig. 3) because the organic molecules move to the surface of the particle (Fig. 4a–b). The fraction of CPA molecules in the outermost

region of the particle covering 5 % of its total volume increases from 65 to 82 % as the RH increases. The accumulation of CPA molecules at the particle surface is further reflected in the corresponding density profiles inside the nanoparticle (Fig. S5). Also, the probability of finding two CPA molecules next to each other is higher at higher RH levels (Fig. S6). CPA is a known surface–active molecule and our

findings are consistent with previous studies (Karadima et al., 2017; Li et al., 2010).

For RH greater than 56 %, the particle core contained only ions and water (Fig. S7). This was also true for the intermediate region for RH higher than 76 %. The fraction of water molecules found in the two innermost domains increased as the RH increased (Fig. S7, S8a). According to our simulations, as the water content increases,

ions leave the particle surface showing a clear preference for the inner regions (Fig.





S7, S8b). Also, as the humidity increases, the ratio of sulfate ions to ammonium ions locally diverges from the nominal value of 2 (Fig. S7). At 88 % RH, there is practically no sulfate at the particle surface but there is a little ammonium present. This result is consistent with the MD study of Gopalakrishnan et al. (2005) for an

5 ammonium sulfate solution, who reported that ammonium ions are more likely to reside near the interface than sulfate ones.

Our results suggest that as the RH increases, the separation between organic molecules and ions becomes more pronounced. The region near the surface accommodates practically all CPA molecules whereas the ions prefer to reside in the

10 interior domains (the core and the ring). The CPA molecules are either alone or in the form of small islands composed of 2–3 molecules at the surface, although these formations change during the simulation.

### 3.2.2 Particles containing MBTCA and *n*-alkanes

Particles containing MBTCA, *n*-eicosane and *n*-triacontane (7–14 %) were

15 also simulated at high RH (88 %). In our simulations, the formed particles contained 10 organic molecules (MBTCA, *n*-eicosane or *n*-triacontane), 3200 water molecules, 200 sulfate and 400 ammonium ions (simulations No 10–12 in Table 1). The resulting particles were almost spherical with an equivalent diameter of about 6 nm, and a mass density around 1.3 g cm$^{-3}$.

A separation between organics and ions was again observed, exactly as in the case of the CPA–containing particles. Organic molecules resided at the surface of the particle either as isolated entities (the MBTCA) or organized in small structures (the normal alkanes) (Fig. 4c–f, S9). In contrast to the CPA–containing particles, no MBTCA islands were observed (Fig. 4c–d). MBTCA contains a shorter alkyl group

compared to CPA, plus three carboxyl groups that can easily form hydrogen bonds with the water molecules. On the other hand, alkane molecules were found to prefer to stay close to each other (due to their hydrophobicity) in order to minimize contact with water molecules (Fig. 4e–f, S9). The alkane molecules form a surface–lens. Ions are mainly located inside the particle and not at its surface, while water molecules are

found everywhere.



The inner particle regions were depleted of organic molecules (Fig. 4c–f, S9–11); the distribution of inorganic species, on the other hand, was similar for all nanoparticles (Fig. S11). The water content was found to increase gradually and smoothly from the core to the surface for all particles (Fig. S12a). The gradient of the water profile is related to ion competition for hydration in the bulk, where they dissolve in the water present in the particle from the outer to the inner regions. Sulfate anions could hardly be found at the surface (Figs. 12b and S11), showing a clear preference for the inner domains of the particle (especially the core). However, some ammonium cations (which in all cases examined here tended to follow the sulfate anions) could be found at the surface (Fig. S11).

### 3.3 Effect of organic mass

In all previous cases, organics represented less than 15 % of the dry particle mass, the latter being dominated by ammonium and sulfate ions. In this section, we discuss the effect of increasing organic content on particle morphology and structure, at low or intermediate RH conditions.

### 3.3.1 CPA–containing nanoparticles

Nanoparticles composed of 100, 200 and 400 CPA molecules (together with 200 sulfate and 400 ammonium ions, and 400 water molecules) were simulated. The organic mass fraction of the formed nanoparticles ranged from 7 to 74 % dry (simulations No 1 and 13–15 in Table 1). The particle size increased, and the largest nanoparticle had an equivalent diameter of about 7 nm. The density decreased from 1.4 to 1.1 g cm$^{-3}$ (Table 1).

The percentage of CPA molecules at the particle core increased with increasing organic mass fraction (Fig. 5), but most of the CPA was located at the surface, which caused a shift in the distribution of water molecules towards the interior of the particle (Fig. S13 and 14a) and had only a small effect on the distribution of ions (Fig. S14b). The distribution of water molecules became more uniform as the CPA mass fraction increased. Also, for all organic mass fractions considered, the majority of ions were located at the two inner regions (Fig. S14b).



According to Fig. 6, aerosol particles with a high organic content can be non–spherical and inhomogeneous. In these particles, CPA molecules prefer to form independent neighbourhoods. Phase separation was again observed, with organic molecules covering the surface of the particle and ions and water molecules residing

at the inner domains. The formed nanoparticles are highly inhomogeneous, consisting of an organic layer covering a complex inorganic phase containing also organic inclusions. Increasing the organic mass content forced CPA molecules to move deeper in the nanoparticle (Fig. 6b).

### 3.3.2 MBTCA–containing nanoparticles

The nanoparticle examined here consisted of 100 MBTCA molecules, 200 sulfate and 400 ammonium ions and 400 water molecules. It was characterized by a dry organic mass fraction of 43 %, an equivalent diameter around 5.1 nm, and a density equal to 1.3 g cm$^{-3}$ (simulation No 16 in Table 1). Increasing the MBTCA content from 7 to 43 % had a small effect on its distribution; there was a small increase in the number of

MBTCA and water molecules at the core of the particle (Fig. S15). The 43 % MBTCA–containing particle was highly inhomogeneous (Fig. S16), with MBTCA molecules residing also inside the particle. This structure is different from that exhibited by the CPA–containing particle at similar compositions, wherein such formations were observed at higher organic mass fractions (58 and 74 %).

### 3.3.3 Eicosane–containing nanoparticle

We also examined the effect of increased water content (68 % RH) in a nanoparticle composed mainly of $n$-eicosane (63 % dry mass). The simulated particle (200 $n$-eicosane molecules, 200 sulfate and 400 ammonium ions, 1600 water molecules) had an equivalent diameter of 7.4 nm (simulation 17 in Table 1).

Eicosane molecules formed a well–organized structure (Fig. 7) characterized by close packing of the eicosane chains in the form of lamellae extending from the core of the nanoparticle to its outer surface along different directions (Fig. 7a and Fig. S17). The particle consisted of a continuous organic phase and several separate inorganic/aqueous domains (Fig. 7), resulting in a highly inhomogeneous structure.

Alkane molecules could be found everywhere in the particle, which should be



contrasted with the case of the eicosane particle at low organic mass fraction and low humidity levels (Fig. S18a). Most of the ions were at the surface and not inside the particle (Fig. S18c).

### 3.4 Effect of increasing relative humidity at high organic (CPA) mass

Nanoparticles with 74 % dry w/w CPA content were simulated at 20, 60 and 80 % RH (simulations No 15, 18 and 19 in Table 1). Their equivalent diameters increased from about 6.8 nm at 20 % RH, to about 6.9 nm at 60 % RH, and to about 7.5 nm at 80 % RH. The asphericity of the particles decreased as the RH increased, and it practically vanished at 80 % RH (Table 1).

The effect of an enhanced RH in these particles at high CPA content was similar to that at low CPA content. The presence of CPA molecules at the surface of the particle was enhanced as the humidity increased. For example, at 80 % RH, 70 % of the CPA was at the surface of the particle (Fig. 8). In contrast, the concentration of organics in the core was almost zero at 80 % RH. We also observed the complete
absence of ions from the surface of the particle and the low presence of water molecules there (Fig. 8, 9b, d, S19).

    The preference of CPA molecules for the surface of the particle leads to the formation of an organic layer (a coating), resulting in a core–shell morphology as the humidity increases, driven by the salting–out effect (Fig. 9a, c and S19).

### 3.5 Particles containing multiple organic components

We proceed now to the analysis of the simulation results for particles composed of a mixture of different organic compounds.

### 3.5.1 CPA and MBTCA mixtures

Particles composed of CPA, MBTCA, sulfate, ammonium and water were simulated
(simulations No 20–23 in Table 1) at different dry organic content (43 and 75 %) and RH (31–86 %). The nanoparticles had an equivalent diameter between 5.2 and 7.5 nm, and their density ranged from 1.2 to 1.3 g cm$^{-3}$. The particles at higher RH were quite spherical (Table 1).





In these particles, the increase of the humidity and of the organic mass fraction resulted in a separation of the organic compounds, with the CPA molecules partially covering the MBTCA ones (Fig. 10a–b, d–e). This was less evident in the case of the particle with 43 % organic mass at low RH (Fig. S20a–b) where both types of

organics were present at the surface (Fig. S21). However, at a higher RH of 86 % (keeping the organic mass content constant), CPA and MBTCA were concentrated even more at the surface of the particle, with CPA molecules residing mostly at the outermost areas and many MBTCA molecules lying just underneath (Fig. S20d–e). At 75 % organic mass fraction and 67 % RH, the layering between CPA and MBTCA

molecules was more pronounced, with the majority of MBTCA residing now inside the particle (Fig. 10a–b, S21). Such a layered arrangement of CPA and MBTCA became even more pronounced at 80 % RH (Fig. 11d–e, S21). The ratio of CPA to MBTCA molecules present in the very outer region of the particle representing 5 % of its total volume increased gradually from 1 to 3 as the organic content and the relative

humidity increased (simulations No 20–23). Given that CPA is a surfactant, the separation of CPA and MBTCA molecules at these conditions implies that CPA hinders MBTCA molecules from reaching the surface of the particle. In the particles examined here, the MBTCA fraction at the interior of the nanoparticles was greater (up to 3 times, at 75 % organic mass and 80 % RH) than the CPA fraction (Fig. S21).

Therefore, at least 70 % of the CPA was at the surface of the particle, while MBTCA was always present at the core (Fig. S21).

According to our simulations, increasing the organic mass fraction and especially the RH results in the formation of a core–shell structure as organics, especially CPA, are salted–out by ions. The surface is enriched in organics, and the

core in inorganics (Fig. S22–23). For example, for the particle with 75 % organic mass fraction at 67 % RH, less than 10 % of the water molecules were at the surface; ions, on the other hand, were found in the two interior regions, with more than 70 % of them located at the core (Fig. S22–23). The intermediate ring–like particle region plays therefore the role of a transition area from an organic–rich phase to an

inorganic–rich one. The coating created by the organics at the surface of the particle is not uniform as it is composed of several small organic neighbourhoods (in a structure



that resembles a random network) (Fig. 11a–b, d–e, S20a–b, d–e). The inorganics at the interior of the particle, on the other hand, form either inter–connected inclusions (Fig. 11c, S20c) or a continuous phase (Fig. 11f, S20f), depending on the prevailing RH conditions.

5 **3.5.2 CPA and triacontane mixtures**

We also simulated nanoparticles consisting of CPA and *n*-triacontane (simulations No 24–26), plus the inorganic ions. These had a density between 1.0 and 1.2 g cm$^{-3}$ and equivalent diameters that ranged from 5.6 to 8 nm. The particle with and 82 % organic mass fraction at 18 % RH had the highest asphericity among all nanoparticles 10 examined in our work (Table 1).

The resulting nanoparticles included multiple hydrophobic and hydrophilic domains; the alkanes formed the hydrophobic region, while the first hydrophilic domain contained the inorganic ions and the second mainly CPA (Fig. 11). At 86 % RH, the alkanes formed a cap over the almost spherical aqueous solution. Most of the 15 CPA molecules were at the area between the alkanes and the aqueous drop (Fig. 11a–c). At low RH, the inorganic domain consisted of a few inter–connected (Fig. 11f) or not connected (Fig. 11i) regions. The alkane domain also included several connected structures (Figs. 11e-h and S24). The CPA molecules were either at the surface covering the aqueous regions or between the alkanes and these hydrophilic domains 20 (Fig. 11d, g). In all nanoparticles discussed in this section, a lot of CPA was at the interface between hydrophobic and aqueous regions, with their methyl (hydrophobic) groups pointing towards the alkane phase. The fraction of CPA molecules at the surface of the particles was lower than that observed in the CPA–containing particles because of the simultaneous presence of *n*-triacontane and its self–assembly. As 25 expected, due to the enhanced organic content, organic species were present almost everywhere in the particle.



### 3.6 Sensitivity simulations

#### 3.6.1 Sensitivity to temperature

Additional simulations were performed to examine the effect of temperature on the resulting morphologies. We repeated the simulation of the CPA–containing particle with 7 % dry organic mass content at 40 % RH (simulation No 1) at two lower temperatures, 286 and 300 K. The nanoparticle with 10 % *n*-eicosane mass fraction at 40 % RH (simulation No 3) was also simulated at 300 K. Finally, the simulation of the particle with 74 % dry CPA mass content at 20 % RH (simulation No 15) was also repeated at 350 K.

The resulting morphologies in all cases were quite similar to those already reported and discussed in the previous sections. The predictions for the key properties (e.g., the distribution of the constituent species inside the particles) were in all cases within the expected statistical uncertainty (Karadima et al., 2017).

#### 3.6.2 Sensitivity to initial system configuration and simulation length

We also repeated all simulations listed in Table 1 except No 4, 10–12, 19, 20–21 and 25 under the same temperature and pressure conditions but starting from different initial configurations for the corresponding systems. Again, the final morphologies and the predictions obtained were within the expected statistical uncertainty of the results already discussed.

Moreover, we extended the simulation time for systems 1, 5, 8, 13, 14 and 24 to 100 ns. Again, the longer simulation duration did not affect either the final structures or the predicted numerical values of the physical properties of interest.

### 3.7 Summary of particle morphologies

The relative humidity, the type of organic compound present and the organic mass fraction all affect the morphology of the nanoparticles formed during the simulation. Organic-inorganic phase separation was observed in almost all cases. Organic molecules were found at or near the surface of the particle, in contrast to inorganic ions which concentrated mainly at the interior of the particle. Hydrophilic organic molecules associate more with other organics than with water, thus the frequently



assumed existence of a single aqueous phase containing both inorganic ions and secondary organics is not supported by the results of our simulations. Even more, a separation of the organic molecules appears possible: our simulations showed that long–chain alkanes segregate from CPA molecules and that CPA tends to reside in

different areas of the particles than MBTCA. The three general types of nanoparticle morphologies predicted by our simulations are summarized in Table 2.

In the first type, the organics mainly form isolated islands at the surface of the particle that has an inorganic core. Some organic molecules may exist further inside the particle. This type is found in particles containing a low or moderate amount of

oxidized (secondary) organic components (dry mass fraction less than 40 %) at low to moderate RH (less than 55 %).

The second type corresponds to different forms of a core-shell morphology. The core can be multiple (IIa) or one (IIb) inorganic regions and the shell consists of secondary organic components covering the inorganics. The organics are salted–out

by the inorganics with the latter dominating the inner domains of the particle. This type is predicted for particles in which the oxidized organics represent more than 40 % of the dry mass of the particle at a range of RH values as low as 20 %. Multiple inorganic inclusions are predicted at low and moderate RH (20-70 %), while a single core is present at higher RH or higher organic contents. In the case of a mixture of

CPA and MBTCA molecules, CPA molecules are predicted to accumulate at the surface of the particle with MBTCA ones lying underneath.

The third and more complex morphology is predicted for particles that contain large linear alkanes, inorganic components and potentially oxidized organics. In all cases the alkanes are organized in one or more hydrophobic neighbourhoods and

remain well–separated from the rest of the particle constituents. At low organic mass fractions, regardless of RH, a surface lens is formed by the linear alkanes (type IIIa). For organic dry mass fraction greater than 0.5 and at low or moderate RH, several hydrophobic alkane lamellae with hydrophilic inclusions develop (type IIIb). In all these cases the oxidized organics are found either between the alkanes and the

aqueous solution or at the surface of the aqueous solution.



## 3.8 Comparison with experimental results

Phase separation of organics and inorganics but also between hydrophobic and hydrophilic organics was predicted for almost all particles simulated here. This behavior is consistent with the observations of Reid et al. (2011) who reported the
formation of distinct hydrophilic and hydrophobic domains for *n*-decane/aqueous sodium chloride particles at high RH. Song et al. (2007) also reported that *α*-pinene products do not mix well with hydrophobic organic aerosol. Phase separation has been also observed by Marcolli and Krieger (2006) who examined 1,2-hexanediol (O:C ratio = 0.33, close to the O:C ratio of CPA) with several salt aqueous solutions. You
et al. (2014) concluded that phase separation is always observed for O:C < 0.56 and is organic molecule–dependent for 0.56 < O:C < 0.8. This is also consistent with the results of our MD study for the three O:C values (0, 0.3, 0.75) studied in these simulations. On the other hand, in all of our simulations the oxygenated organics were salted–out by the inorganics, while Veghte et al. (2014) reported a homogeneous
morphology for particles with diameters less than 270 nm. In that study, dry particles were investigated.

## 4 Conclusions and atmospheric implications

We investigated the morphology of multicomponent atmospheric nanoparticles composed of sulfate and ammonium ions, water and CPA, MBTCA or
normal alkanes with atomistic MD simulations. A variety of microstructures was observed, depending on composition, humidity and organic mass fraction assumed in the simulation. Three main particle types were identified: organic islands at the surface, inorganic core-organic shell morphologies, and complex structures with hydrophobic and hydrophilic domains. The oxidized organics form the first two types
while in particles with hydrophobic organics the third type is prevalent.

The surface–active CPA molecules were mainly located at the outer regions of the particles. As the water content increased, the CPA molecules were found even closer to the surface, with the interior of the particle composed exclusively of inorganic species. The highly oxygenated MBTCA was more likely to be found at the
particle core compared to CPA. When MBTCA coexisted with CPA, CPA created a



thin layer at the surface, thus providing a shield for MBTCA molecules that resided underneath.

The relatively large alkanes (*n*-eicosane, *n*-tetracosane, and *n*-triacontane) were found to form well–organized structures at the particle surface due to their
strong hydrophobicity. These could extend, depending on the chain length of the alkane, to the inner regions, forming a single or several lamella configurations for low or moderate organic mass fractions, respectively. The hydrophobic alkanes separated not only from ions and water but also from other oxygenated organic species. In particles where both alkanes and oxidized organics were present, the oxidized organic
was located mainly in the zones between the hydrophobic and aqueous domains of the particle.

Increasing the RH resulted in surface enrichment of organics for CPA and/or MBTCA–containing particles. Increasing the organic mass fraction enhanced the presence of organics throughout the particle; however, this effect was also strongly
RH–dependent. As the RH increased, the oxidized organics were salted–out by the ions and concentrated at the surface, while ions accumulated at the core. Depending on concentration, the ions formed inclusions or were distributed quite homogeneously in the core. The hydrophobic alkanes formed separate structures. High humidity levels resulted in the formation of a unique hydrophobic alkane domain (lens–like
configuration).

The nanoparticles studied here were highly heterogeneous, characterized by a small degree of mixing. Therefore, the assumption of a well–mixed state adopted by some chemical transport models is not corroborated by our results.

*Data availability*. The data in the study are available from the authors upon request
(kkaradima@chemeng.upatras.gr).

*Author contributions.* KSK conducted the simulations, analysed the results, and wrote the paper. VGM contributed to the design of the study, the analysis of the results and the writing of the paper. SNP was responsible for the design and coordination of the
study and the synthesis of the results.




*Competing interests*. The authors declare that they have no conflict of interest.

*Acknowledgements*. This study was financially supported by the European Union's Horizon 2020 EUROCHAMP–2020 Infrastructure Activity (Grant agreement 730997) and the U.S. National Science Foundation (Grant agreement 1455244). The authors

acknowledge the computational time granted by the Greek Research & Technology Network (GRNET) in the National HPC facility–ARIS under the project name AtmoStruc (pr004005).

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





**Table 1.** Parameters and particle characteristics for the MD simulations.

| No | Number and type of organic molecules | Number of water molecules (Total/In the particle) | Organic mass (%) | Dry organic mass (%) | RH (%) | Particle density (g cm$^{-3}$) | $D_{eq}$ (nm) | $\kappa^2$ |
|---|---|---|---|---|---|---|---|---|
| 1 | 10 CPA | 400/399 | 5 | 7 | 40 | 1.4 | 4.3 | 0.07 |
| 2 | 10 MBTCA | 400/398 | 6 | 7 | 40 | 1.4 | 4.3 | 0.13 |
| 3 | 10 n-eicosane | 400/399 | 8 | 10 | 40 | 1.4 | 4.3 | 0.03 |
| 4 | 10 n-tetracosane | 400/399 | 9 | 11 | 40 | 1.4 | 4.3 | 0.34 |
| 5 | 10 n-triantacontane | 400/400 | 11 | 14 | 40 | 1.4 | 4.4 | 0.19 |
| 6 | 10 CPA | 800/793 | 4 | 7 | 56 | 1.5 | 4.5 | 0.01 |
| 7 | 10 CPA | 1200/1191 | 4 | 7 | 68 | 1.5 | 4.8 | 0.07 |
| 8 | 10 CPA | 1600/1590 | 3 | 7 | 76 | 1.4 | 5.0 | 0.01 |
| 9 | 10 CPA | 3200/3190 | 2 | 7 | 88 | 1.3 | 5.9 | 0.00 |
| 10 | 10 MBTCA | 3200/3190 | 2 | 7 | 88 | 1.3 | 5.9 | 0.00 |
| 11 | 10 n-eicosane | 3200/3190 | 3 | 10 | 88 | 1.3 | 6.0 | 0.01 |
| 12 | 10 n-triantacontane | 3200/3190 | 5 | 14 | 88 | 1.3 | 6.1 | 0.01 |
| 13 | 100 CPA | 400/398 | 35 | 41 | 32 | 1.4 | 4.9 | 0.16 |
| 14 | 200 CPA | 400/398 | 52 | 58 | 26 | 1.3 | 5.6 | 0.09 |
| 15 | 400 CPA | 400/393 | 69 | 74 | 20 | 1.1 | 6.8 | 0.09 |
| 16 | 100 MBTCA | 400/398 | 37 | 43 | 32 | 1.3 | 5.1 | 0.13 |
| 17 | 200 n-eicosane | 1600/1590 | 68 | 63 | 68 | 0.9 | 7.4 | 0.11 |
| 18 | 400 CPA | 1600/1585 | 57 | 74 | 60 | 1.2 | 6.9 | 0.01 |
| 19 | 400 CPA | 3200/3180 | 47 | 74 | 80 | 1.2 | 7.5 | 0.00 |
| 20 | 50 CPA-50 MBTCA | 400/398 | 36 | 43 | 31 | 1.2 | 5.2 | 0.27 |
| 21 | 50 CPA-50 MBTCA | 3200/3190 | 19 | 43 | 86 | 1.3 | 6.4 | 0.00 |
| 22 | 200 CPA-200 MBTCA | 1600/1582 | 59 | 75 | 67 | 1.3 | 6.9 | 0.01 |
| 23 | 200 CPA-200 MBTCA | 3200/3185 | 48 | 75 | 80 | 1.2 | 7.5 | 0.00 |
| 24 | 50 CPA-50 n-triantacontane | 3200/3176 | 27 | 53 | 86 | 1.2 | 6.8 | 0.05 |
| 25 | 50 CPA-50 n-triacontane | 400/399 | 47 | 53 | 30 | 1.1 | 5.6 | 0.05 |
| 26 | 200 CPA-200 n-triantacontane | 400/397 | 79 | 82 | 18 | 1.0 | 8.0 | 0.21 |





**Table 2.** Description and schematic illustration of the morphologies identified in the simulated atmospheric nanoparticles.

| Type | Description | Illustration[1] | OA/PM[2] | % RH | Simulation |
|------|-------------|-----------------|----------|------|------------|
| **I** | Organic–enriched surface with islands | | $\leq 0.4$<br>$< 0.1$ | $\leq 40$<br>$\leq 55$ | 1, 2, 13<br>6, 7, 8, 9, 10 |
| **IIa** | Core-shell; organic surface with inorganic inclusions | | $> 0.4$ | $< 70$ | 14–16, 20, 22 |
| **IIb** | Core–shell; organic surface with inorganic core | | $> 0.7$<br>$> 0.4$ | $\geq 60$<br>$> 80$ | 18, 19, 23<br>21 |
| **IIIa** | Hydrophobic lens with hydrophilic body | | $< 0.2$<br>$< 0.6$ | $\leq 40$<br>$> 80$ | 3, 4, 5, 11, 12<br>24 |
| **IIIb** | Hydrophobic phase with hydrophilic inclusions | | $> 0.5$ | $< 70$ | 17, 25, 26 |

[1]Colour notation: Hydrophilic organics (green), hydrophobic organics (green leaf), inorganics (blue).

[2]OA/PM: Organic Aerosol/Particulate Matter (dry organic mass fraction)





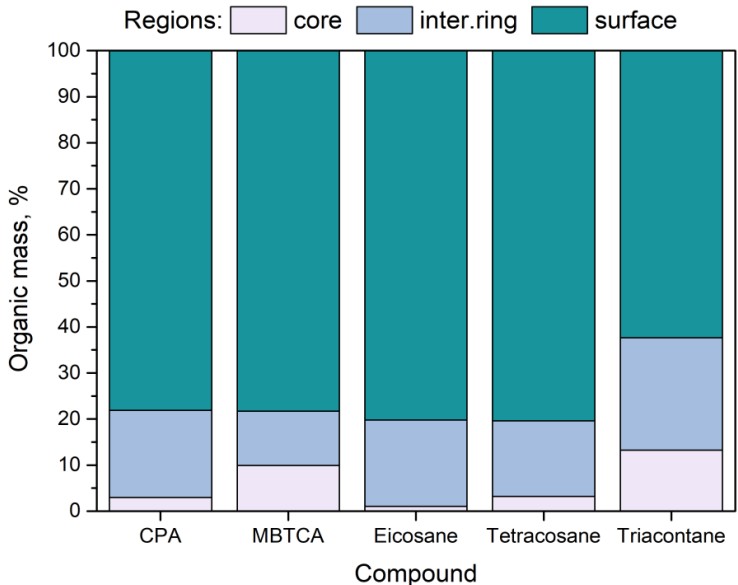

**Figure 1.** Percentage of organic compounds (CPA, MBTCA, *n*-eicosane, *n*-tetracosane and *n*-triacontane) in the three regions (core, intermediate ring, outer surface) of the simulated nanoparticles. Conditions: low organic mass fraction (7–14 % dry) and approximately 40 % RH.





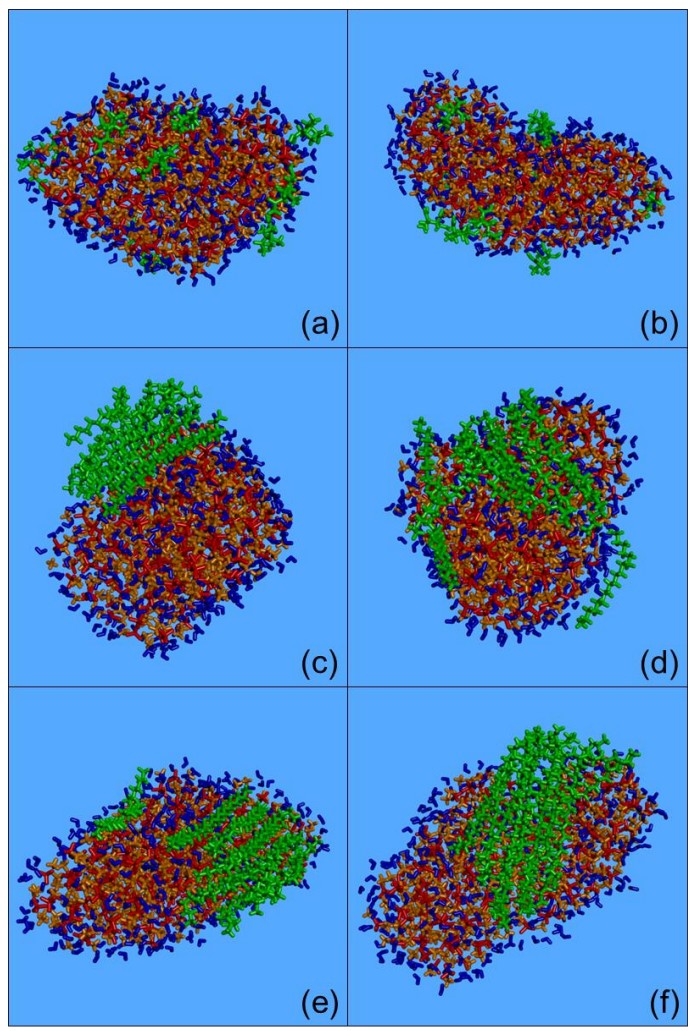

**Figure 2.** Typical snapshots of the nanoparticles composed of (green colour): (a) CPA, (b) MBTCA, (c–d) *n*-eicosane, (e) *n*-tetracosane, and (f) *n*-triacontane, and sulfate anions (red), ammonium cations (orange), and water molecules (blue) from the MD simulations. Conditions: low organic mass fraction (7–14 % dry) and 40 % RH.




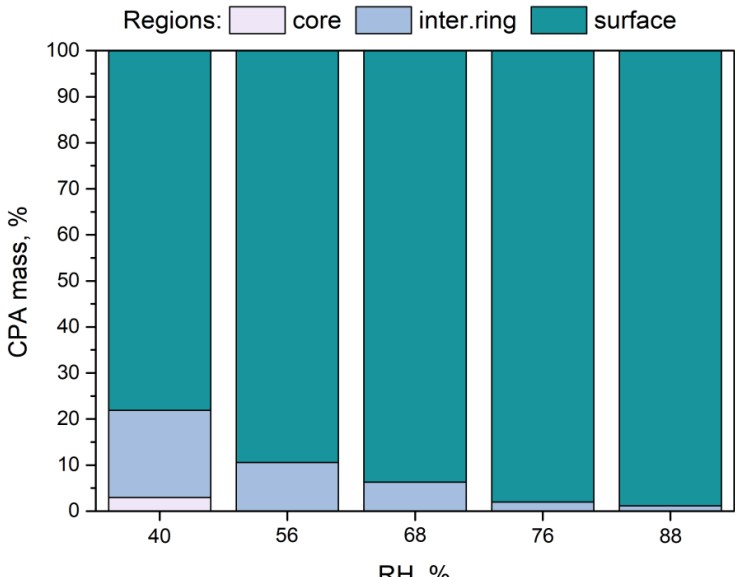

**Figure 3.** Percentage of CPA molecules in the three regions (core, intermediate ring,
5   outer surface) of the simulated nanoparticles with low organic mass content (7 % dry),
as a function of RH.





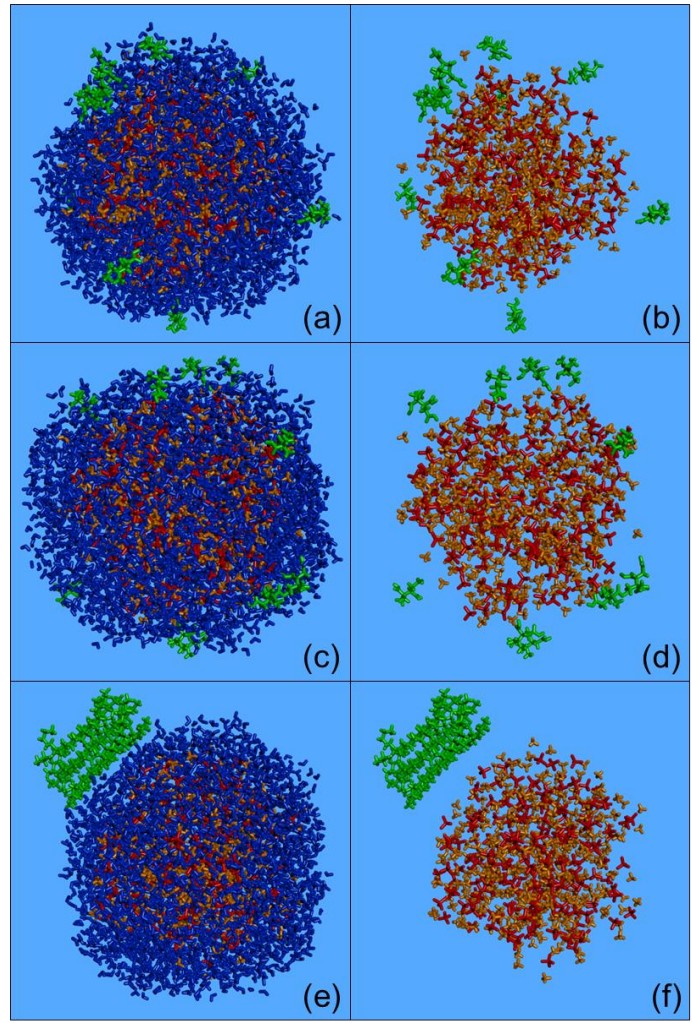

**Figure 4.** Characteristic snapshots from the MD simulations of the nanoparticles containing CPA (a–b), MBTCA (c–d), and *n*-eicosane (e–f) at low organic mass fraction (7–10 %) and 88 % RH, including (a, c, e) and omitting (b, d, f) water molecules. Colour coding: green for CPA, MBTCA, and *n*-eicosane, red for sulfate, orange for ammonium, and blue for water.



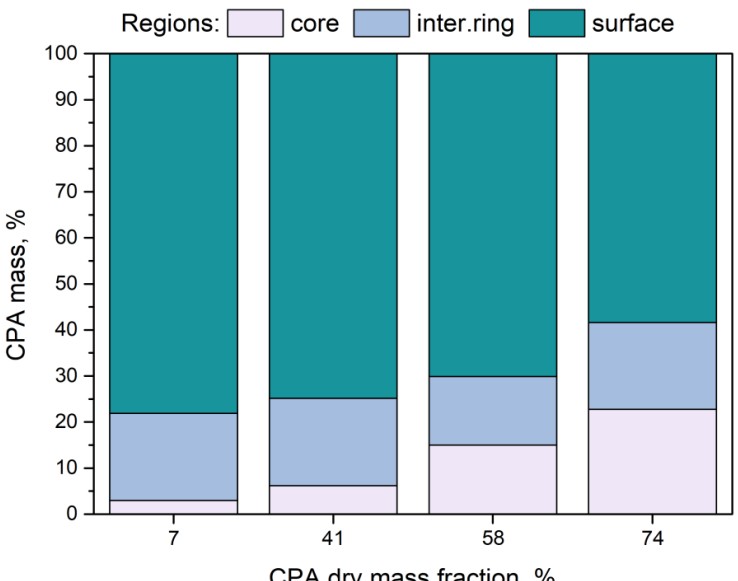

**Figure 5.** Percentage of CPA molecules in the three regions (core, intermediate ring, outer surface) of the simulated nanoparticles as a function of dry organic mass fraction at low RH conditions.



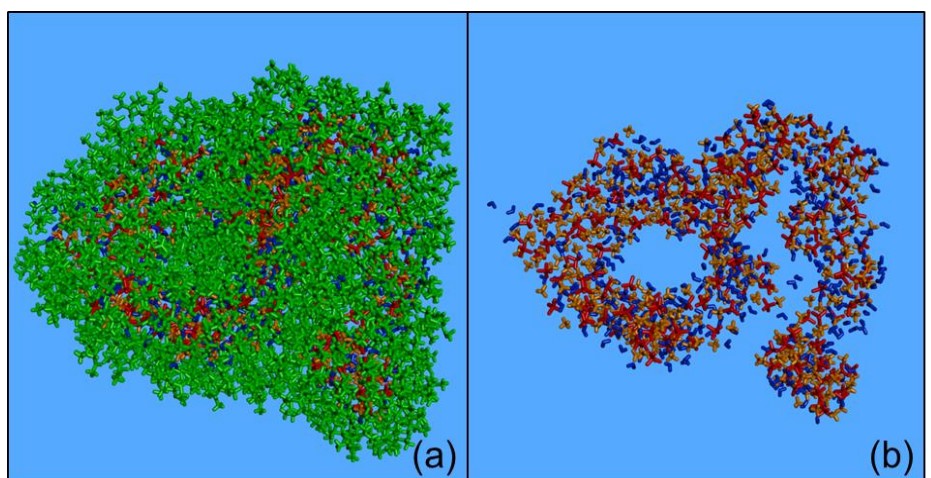

**Figure 6.** Characteristic snapshot from the MD simulation of the nanoparticle
5   containing CPA (74 % dry) at 20 % RH, (a) including and (b) omitting CPA
molecules. Colour coding: green for CPA, red for sulfate, orange for ammonium, and
blue for water.





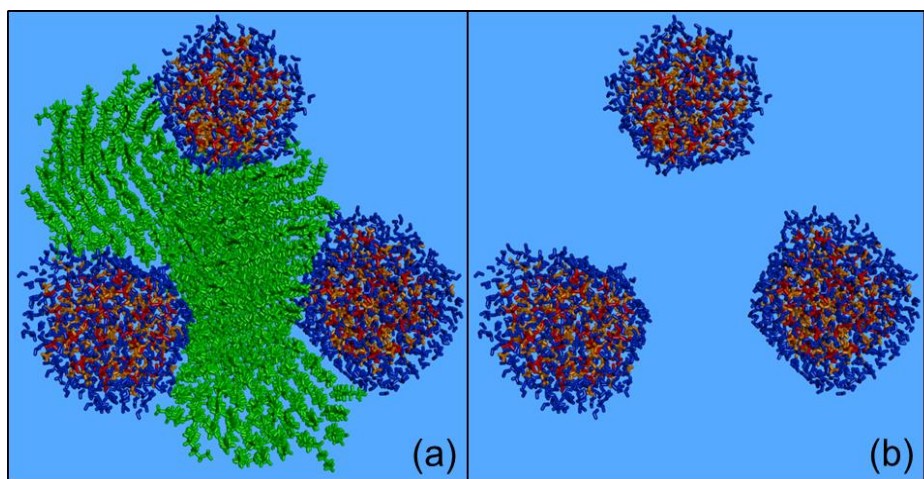

**Figure 7.** Characteristic snapshot from the MD simulation of the nanoparticle
containing *n*-eicosane (63 % dry) at 68 % RH, (a) including (a) and (b) omitting *n*-eicosane molecules. Colour coding: green for *n*-eicosane, red for sulfate, orange for ammonium, and blue for water.





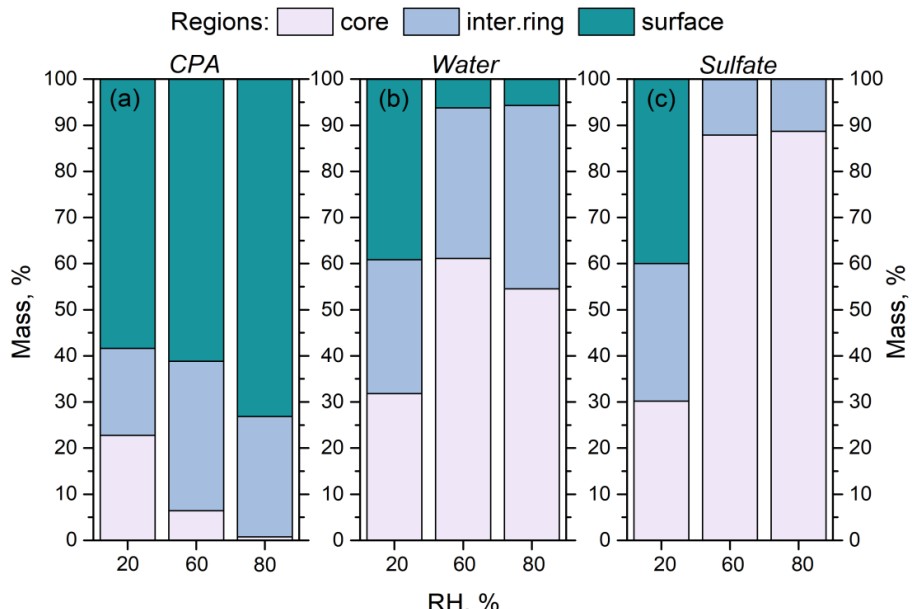

**Figure 8.** Percentage of: (a) CPA molecules, (b) water molecules and (c) sulfate ions in the three regions (core, intermediate ring, outer surface) of the simulated nanoparticles, as a function of RH. The dry organic mass content is 74 %.





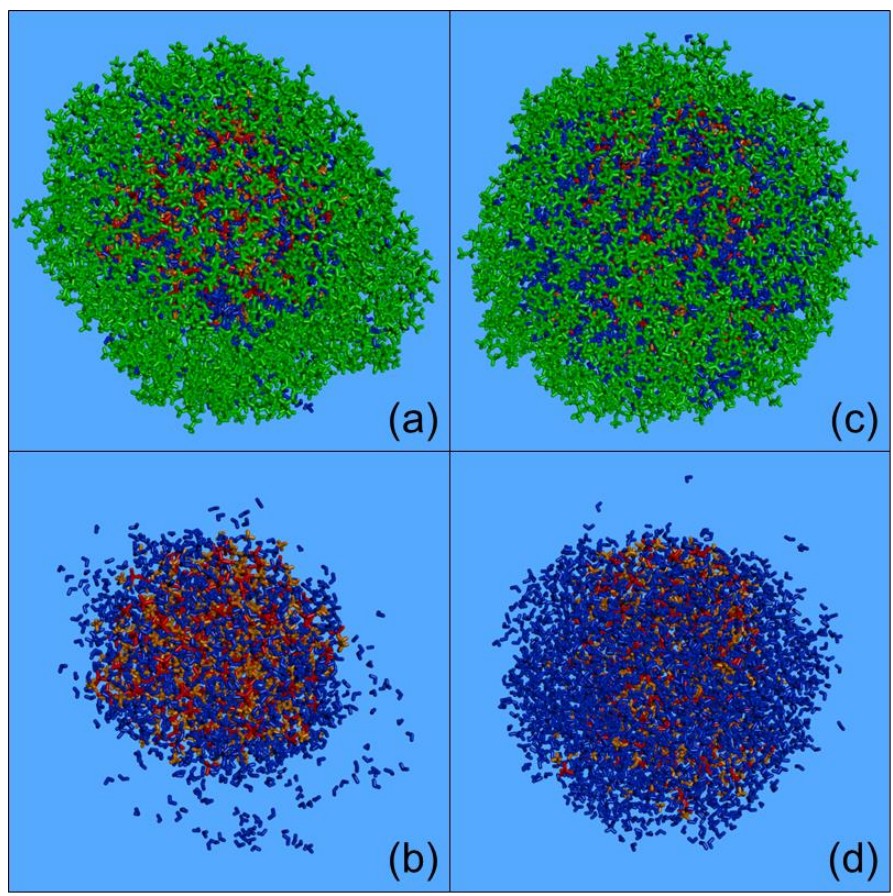

**Figure 9.** Characteristic snapshots from the MD simulation of the nanoparticles containing CPA (74 % dry): (a) at 60 % RH including CPA molecules, (b) at 60 % RH omitting CPA molecules, (c) at 80 % RH including CPA molecules, and (d) at 80 % RH omitting CPA molecules. Colour coding: green for CPA, red for sulfate, orange for ammonium, and blue for water.



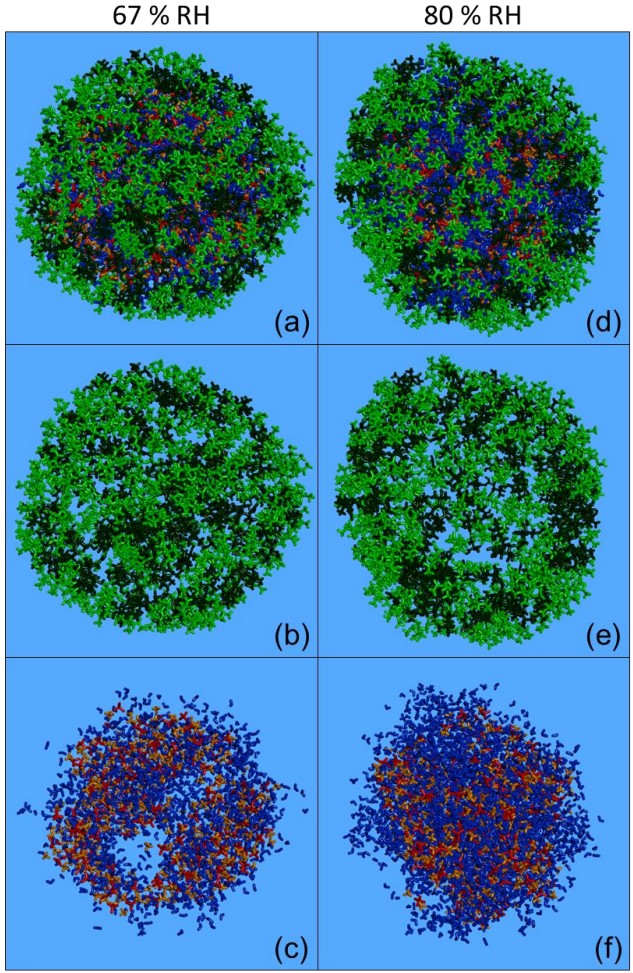

**Figure 10.** Characteristic snapshots from the MD simulations of the nanoparticles containing (dry mass fraction) 36 % CPA and 39 % MBTCA (a) at 67 % RH, (b) at 67 % RH including only organic molecules, (c) at 67 % RH omitting organic molecules, (d) at 80 % RH, (e) at 80 % RH including only organic molecules, and (f) at 80 % RH omitting organic molecules. Colour coding: green for CPA, dark green for MBTCA, red for sulfate, orange for ammonium, and blue for water.



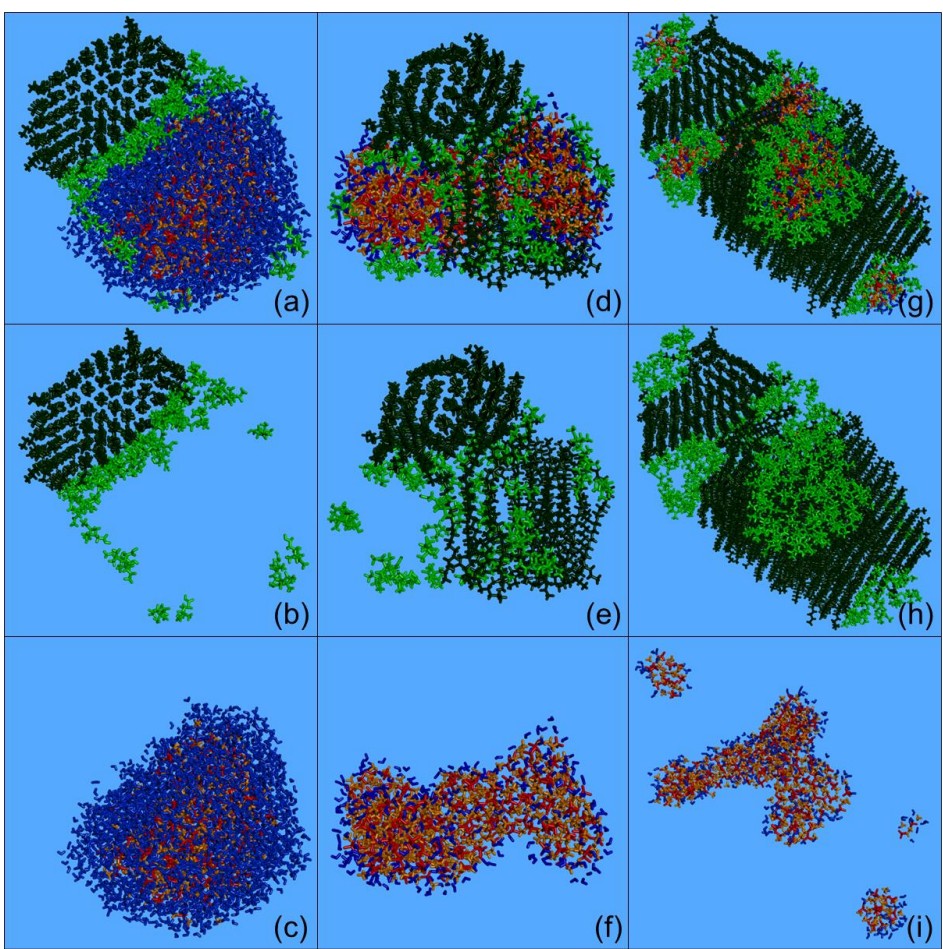

**Figure 11.** Characteristic snapshots from the MD simulations of the nanoparticle containing CPA and *n*-triacontane: (a) at 86 % RH (53 % dry organic content), (b) at 86 % RH (53 % organics) showing only organic molecules, (c) at 86 % RH (53 % organics) omitting organic molecules, (d) at 30 % RH (53 % dry organic content), (e) at 30 % RH (53 % organics) including only organic molecules, (f) at 30 % RH (53 % organics) omitting organic molecules, (g) at 18 % RH (82 % dry organic content), (h) at 18 % RH (82 % organics) including only organic molecules, and (i) at 18 % RH (82 % organics) omitting organic molecules. Colour coding: green for CPA, dark green for *n*-triacontane, red for sulfate, orange for ammonium, and blue for water.