# Peer review of "Insights into the morphology of multicomponent organic/inorganic aerosols from molecular dynamics simulations"

_Atmospheric Chemistry and Physics, 2018_

## Referee Comment (RC1) · Anonymous Referee #1 · 30 Nov 2018

**1   Synopsis**

The topics of particle morphology, salting-out effects of ions on organic compounds and liquid–liquid phase separation in multicomponent aerosol particles are of great interest in the field of aerosol science and atmospheric chemistry and physics. Karadima et al. performed a series of molecular dynamics simulations of nanometer-sized particles using ammonium and sulfate ions as well as water molecules as inorganic components combined with organic components of different functionalities and hydrophilicities. They describe the observed morphologies, including the prevalent feature of phase separation and the propensity of hydrophobic organic compounds to diffuse to

the particle surface layer as a function of system composition at different water contents.

This concise article discusses interesting insights about the internal structures of nanoparticles. It is very well written and accompanied by an appropriate set of high-quality figures and tables. I found only a few minor issues that are suggested to be addressed by the authors. These are listed in the following.

**2  General comments**

- Equilibrium relative humidity (RH) reported is not accounting for size effect.
  On page 8, lines 4, it is stated that the E-AIM model was used to estimate the equilibrium RH for the particles studied. As far as I understand, the reported RH is equivalent to the computed water activity in the particles, which was determined here by the mole fractions of water, ammonium and sulfate ions (were the organics considered in the calculation?). Therefore, the approximate RH values, reported in Table 1 and the text, represent the equilibrium RH of macroscopic (bulk) solutions of the considered compositions, but most likely not that of nanometer-sized particles. At the particle diameter scale of this study, the Kelvin effect is non-negligible (a factor of about 1.3 - 1.6 here scaling the water activity) due the large surface area to volume ratio, which should be considered with the Köhler equation to determine the appropriate equilibrium RH. As a consequence, $\sim 40$ % RH may actually be about $\sim 55$ to 60 % RH in equilibrium with these particles and the higher (bulk equilibrium) RH values reported may represent water vapor supersaturation conditions when the Kelvin effect is considered.
  Ideally, the equilibrium RH would be computed from the simulated gas phase water vapor mixing ratio in the simulation domain; however, I understand that for the chosen small domain size, there is hardly a single water molecule in the gas phase, which renders that approach inadequate. The authors should address
this issue by either adjusting the reported RH or by clearly stating that the values are a reference referring to bulk equilibrium conditions and not to nanoparticles.

- Repeatability of the simulated structures.

Most of the simulation results were obtained by considering a 10 ns time period at the end of a simulation, after allowing particle formation, diffusion and relaxation. While this seems to be an adequate procedure, some of the structures observed, especially cases like the one shown in Fig. 7 or Fig. 11d,g, raise questions about the repeatability of these structures (i) when the initialization of the simulation is changed (which is briefly discussed in Section 3.6.2) or (ii) whether such structures are long-lasting equilibrium or rather transient, metastable configurations. In the case shown in Fig. 7, one could imagine that the true thermodynamic equilibrium configuration (lowest Gibbs energy state) would favor merging of the three distinct aqueous inorganic phases into a single, larger aqueous inorganic phase of smaller surface area, phase-separated from the eicosane-rich phase. Have repeated simulations always resulted in three distance-separated aqueous phases (each of which of similar composition) in the case of this system? Also, would it be possible that the three aqueous ion-rich phases would merge on a substantially longer (but reasonable) time scale, say ms to seconds time rather than the 60 or 100 ns simulated. In other words, how sure are the authors that the simulation results are reflecting stable thermodynamic equilibrium configurations? Considerations of the limited simulation time scale and the procedure followed for initializing the simulations may then also be discussed in the context of the structures IIa vs. IIb shown in Table 2. An extended discussion on such aspects and potential implications for larger particle sizes (tens to hundreds of nm diameter) would be of interest to the community.

**3   Specific comments**

1. Page 2, lines 20 - 22: Consideration of the effect of organic coatings/films/phases on the formation of cloud droplets have also been central topics in key studies by Ruehl et al. (2014; 2016) and Ovadnevaite et al. (2017).

2. Page 3, line 1: "surface partitioning" is bulk–surface partitioning meant?

3. Page 3, lines 12 - 19: In addition to the cited experimental studies, the application of thermodynamic equilibrium models have also provided valuable insights into the factors of organic–inorganic interactions and composition ranges of liquid–liquid phase separation, e.g. the works by Zuend et al. (2010), Zuend and Seinfeld (2012), Renbaum-Wolff et al. (2016), Pye et al. (2018). Conclusions from such studies for the mixing in (non-nanoscale) aerosols are in agreement with the statement made in the last paragraph of Section 4 in this article. The authors may want to extend the discussion on Pages 3 and 21 considering those studies.

4. Page 4, line 2: "theoretical and kinetic models". What do you mean by theoretical; are you referring to thermodynamic models? The latter are not more or less "theoretical" than kinetic models.

5. Page 4, line 29: phrasing: "liquid either glassy"

6. Page 5, line 21: spelling: carboxyl (not carboxylic)

7. Page 16, line 12: I think "Fig. 11d-e" should be Fig. 10d-e.

8. Page 17, lines 1-4: Fig. 11 should be Fig. 10 in all instances there.

9. Page 20, line 9: correct phrasing of "salt aqueous solutions".

10. Page 20, line 15: in this comparison with experimental studies, especially regarding a potential size effect on phase separation, it would be useful to further discuss the simulation results compared to the experimental findings under different particle drying rates by Veghte et al. (2013) and by Altaf and Freedman (2017).

11. Fig. S23 of the Supplement is neither discussed/mentioned in the main text nor the supplement text.

**References**

- Altaf, M. B., and Freedman, M. A.: Effect of Drying Rate on Aerosol Particle Morphology, The Journal of Physical Chemistry Letters, 8, 3613-3618, 10.1021/acs.jpclett.7b01327, 2017.

- Ovadnevaite, J., Zuend, A., Laaksonen, A., Sanchez, K. J., Roberts, G., Ceburnis, D., Decesari, S., Rinaldi, M., Hodas, N., Facchini, M. C., Seinfeld, J. H., and O' Dowd, C.: Surface tension prevails over solute effect in organic-influenced cloud droplet activation, Nature, 546, 637-641, 10.1038/nature22806, 2017.

- Pye, H. O. T., Zuend, A., Fry, J. L., Isaacman-VanWertz, G., Capps, S. L., Appel, K. W., Foroutan, H., Xu, L., Ng, N. L., and Goldstein, A. H.: Coupling of organic and inorganic aerosol systems and the effect on gas–particle partitioning in the southeastern US, Atmos. Chem. Phys., 18, 357-370, 10.5194/acp-18-357-2018, 2018.

- Renbaum-Wolff, L., Song, M., Marcolli, C., Zhang, Y., Liu, P. F., Grayson, J. W., Geiger, F. M., Martin, S. T., and Bertram, A. K.: Observations and implications of liquid–liquid phase separation at high relative humidities in secondary organic material produced by $\alpha$-pinene ozonolysis without inorganic salts, Atmos. Chem. Phys., 16, 7969-7979, 10.5194/acp-16-7969-2016, 2016.

- Ruehl, C. R., and Wilson, K. R.: Surface organic monolayers control the hygroscopic growth of submicrometer particles at high relative humidity, The Journal of Physical Chemistry A, 118, 3952-3966, 2014.

- Ruehl, C. R., Davies, J. F., and Wilson, K. R.: An interfacial mechanism for cloud droplet formation on organic aerosols, Science, 351, 1447-1450, 10.1126/science.aad4889, 2016.

- Veghte, D. P., Altaf, M. B., and Freedman, M. A.: Size Dependence of the Structure of Organic Aerosol, Journal of the American Chemical Society, 135, 16046-16049, 10.1021/ja408903g, 2013.

- Zuend, A., and Seinfeld, J.: Modeling the gas-particle partitioning of secondary organic aerosol: the importance of liquid-liquid phase separation, Atmospheric Chemistry and Physics, 12, 3857-3882, 2012.

- Zuend, A., Marcolli, C., Peter, T., and Seinfeld, J.: Computation of liquid-liquid equilibria and phase stabilities: implications for RH-dependent gas/particle partitioning of organic-inorganic aerosols, Atmospheric Chemistry and Physics, 10, 7795-7820, 2010.

---

## Referee Comment (RC2) · Anonymous Referee #2 · 10 Jan 2019

This manuscript describes the morphology of mixed organic-inorganic nanoparticles under different composition regimes and environmental conditions (particularly RH) according to molecular dynamics simulations. The methods draw heavily upon a previously peer-reviewed paper by the first author (from 2017). The manuscript is written clearly and presents a concise summary of the diverse morphologies that can result from their prescribed conditions. The material is relevant for the ACP community and merits publication after the following comments have been addressed, many of which will likely not affect the conclusions.

For a volume that corresponds to 1 atm and 320K for the system sizes studied, should

there not be about 3 to 9 water vapor molecules per 100 "M" molecules (N2 or O2) for the RHs described?

Is a vapor phase necessary at all? While NPT at 1 atm better reflect experimental and ambient conditions, the barostat used provides isotropic dimension rescaling (i.e., volume rescaling). When simulating a condensed-phase surrounded by "gas", this rescaling should have little effect on the condensed phase as the gas-gas and gas-surface interactions are infrequent over the simulation timescales used in this work (approximately 1 collision per 10 ns?). The condensed-phase properties are therefore maintained primarily by the force field parameterizations, and similar results should be obtained with a NVT simulations with no gas-phase molecules for these "low" pressures (and is more often used in such studies).

Is acid deprotonation neglected in these simulations?

While SPC/E is commonly used, other researchers have previously found polarizable force fields for water to be important for determining distributions of sodium and chloride ions in a system with an air/water interface (Jungwirth and Tobias, 2000). Some mention should probably be made especially for interpreting the morphology from the organic-inorganic-water system.

Jungwirth, P., Tobias, D.J., 2000. Surface Effects on Aqueous Ionic Solvation: A Molecular Dynamics Simulation Study of NaCl at the Air/Water Interface from Infinite Dilution to Saturation. J. Phys. Chem. B 104, 7702–7706. https://doi.org/10.1021/jp000941y

While difficult to generalize these findings at this stage, one conclusion put forth is that the diversity in heterogeneities can result but are currently not well-represented by chemical transport models. On this point, sometimes it is unclear what are surprising findings from this work vs. what was known previously (from experiment or MD simulation). The phase separation of organics by O:C is discussed by the authors, but the dependence of bulk-surface partitioning of mildly soluble species on concentration and composition is of course known (e.g., also demonstrated in molecular simulation

by Hede et al. 2011 cited by the authors), and is anticipated by the Gibbs adsorption isotherm (and more recently by a finite volume model - Malila and Prisle 2018). Phase separation among immiscible organics have been investigated by Ye et al. 2016. Organic islands have been found in experiments (Garland et al. 2008) and in simulation (Hede et al. 2011). Also, aspherical droplets due to water-organic interactions have previously been reported in several papers by Zachariah and co-workers (including that already cited by the authors in the introduction).

Malila, J., & Prisle, N. L. (2018). A monolayer partitioning scheme for droplets of surfactant solutions. Journal of Advances in Modeling Earth Systems, 10. https://doi.org/10.1029/2018MS001456

Garland, E.R., Rosen, E.P., Clarke, L.I., Baer, T., 2008. Structure of submonolayer oleic acid coverages on inorganic aerosol particles: evidence of island formation. Physical Chemistry Chemical Physics 10, 3156–3161. https://doi.org/10.1039/B718013F

Ye, J., Gordon, C.A., Chan, A.W.H., 2016. Enhancement in Secondary Organic Aerosol Formation in the Presence of Preexisting Organic Particle. Environ. Sci. Technol. 50, 3572–3579. https://doi.org/10.1021/acs.est.5b05512

---

## Author Response (AR1)

**Responses to the Comments of Reviewer 1**

**(1)** The topics of particle morphology, salting-out effects of ions on organic compounds and liquid–liquid phase separation in multicomponent aerosol particles are of great interest in the field of aerosol science and atmospheric chemistry and physics. Karadima et al. performed a series of molecular dynamics simulations of nanometer-sized particles using ammonium and sulfate ions as well as water molecules as inorganic components combined with organic components of different functionalities and hydrophilicities. They describe the observed morphologies, including the prevalent feature of phase separation and the propensity of hydrophobic organic compounds to diffuse to the particle surface layer as a function of system composition at different water contents. This concise article discusses interesting insights about the internal structures of nanoparticles. It is very well written and accompanied by an appropriate set of high quality figures and tables. I found only a few minor issues that are suggested to be addressed by the authors. These are listed in the following.

We thank the reviewer for the comments and suggestions. Our responses (in regular font) and corresponding changes to the manuscript (in italics) are given after each comment (in italics).

**General comments**

**(2)** On page 8, lines 4, it is stated that the E-AIM model was used to estimate the equilibrium RH for the particles studied. As far as I understand, the reported RH is equivalent to the computed water activity in the particles, which was determined here by the mole fractions of water, ammonium and sulfate ions (were the organics considered in the calculation?).

The reviewer is correct; the reported RH is indeed equivalent to the predicted water activity in the particles. The values shown now in Tables 1 and 2 do not include the organics in the calculation. This is now explained using a footnote in the tables and the text. We repeated the RH estimation considering the hydrophilic organics in E-AIM assuming an ideal solution. The effect in most cases was a change of the RH by 1-2 percent. The highest effect was at the low RH simulations like Case 20 in which the RH changes from 40 to 31 percent. We have also estimated the RH using the concentration of the water molecules in the gas phase. In all cases this value was just a few percent different than the value based on particle thermodynamics. In some cases the RH calculation based on the number of water molecules is quite uncertain because there are only a few such molecules in the simulation cell. This information has also been added to the paper.

**(3)** Therefore, the approximate RH values, reported in Table 1 and the text, represent the equilibrium RH of macroscopic (bulk) solutions of the considered compositions, but most likely not that of nanometer-sized particles. At the particle diameter scale of this study, the Kelvin effect is non-negligible (a factor of about 1.3 - 1.6 here scaling the water activity) due the large surface area to volume ratio, which should be considered with the Köhler equation to determine the appropriate equilibrium RH. As a consequence, ~ 40 % RH may actually be about ~ 55 to 60 % RH in equilibrium

with these particles and the higher (bulk equilibrium) RH values reported may represent water vapor supersaturation conditions when the Kelvin effect is considered. Ideally, the equilibrium RH would be computed from the simulated gas phase water vapor mixing ratio in the simulation domain; however, I understand that for the chosen small domain size, there is hardly a single water molecule in the gas phase, which renders that approach inadequate. The authors should address this issue by either adjusting the reported RH or by clearly stating that the values are a reference referring to bulk equilibrium conditions and not to nanoparticles.

The reported values refer to the bulk equilibrium conditions and this is now clearly stated in Section 2.3 and Tables 1 and 2. The estimation of the magnitude of the Kelvin effect in these particles is quite uncertain given that in a lot of the investigated systems (e.g., pinonic acid) the organic acts as a surfactant. As a result, the magnitude of the Kelvin effect, depending on the assumed surface tension, can be as high as the reviewer suggests, but also much lower just a few percent. The estimated RH values based on the concentration of the water molecules in the gas phase suggest though that the Kelvin effect (based on the simulation itself) is small and increases the reported values by only a few percent. We now stress this uncertainty of the estimated RH in the revised manuscript. Please note that this uncertainty in the RH does not affect the major conclusions of the paper.

 **(4)** Repeatability of the simulated structures. Most of the simulation results were obtained by considering a 10 ns time period at the end of a simulation, after allowing particle formation, diffusion and relaxation. While this seems to be an adequate procedure, some of the structures observed, especially cases like the one shown in Fig. 7 or Fig. 11d,g, raise questions about the repeatability of these structures (i) when the initialization of the simulation is changed (which is briefly discussed in Section 3.6.2) or (ii) whether such structures are long-lasting equilibrium or rather transient, metastable configurations.

Most of the systems were discussed in the paper were simulated twice. This includes the systems mentioned by the reviewer (shown in Figures 7 and 11). The repeated simulations started from different initial system configurations (different atoms positions and initial velocities). The differences in the final particle morphologies were minor. For example, for the particle shown in Figure 7 (n-eicosane, ammonium sulfate and water), the second simulation resulted once more in three inorganic regions and one extended alkane phase. It is interesting that the two simulations reached this structure following different paths; initially the corresponding particles had a different number of inorganic regions but then these merged to the three regions. For the particles shown in Figure 11d and Figure 11g (n-triancontane, CPA, ammonium sulfate, and water) the final complex structures were also very similar. We have added a detailed discussion of the repeatability of our simulated structures starting from different initial conditions in the revised paper. We have also added pictures of the resulting particles in the SI.

**(5)** In the case shown in Fig. 7, one could imagine that the true thermodynamic equilibrium configuration (lowest Gibbs energy state) would favor merging of the three distinct aqueous inorganic phases into a single, larger aqueous inorganic phase of smaller surface area, phase-separated from the eicosane-rich phase. Have repeated

simulations always resulted in three distance-separated aqueous phases (each of which of similar composition) in the case of this system? Also, would it be possible that the three aqueous ion-rich phases would merge on a substantially longer (but reasonable) time scale, say ms to seconds time rather than the 60 or 100 ns simulated. In other words, how sure are the authors that the simulation results are reflecting stable thermodynamic equilibrium configurations? Considerations of the limited simulation time scale and the procedure followed for initializing the simulations may then also be discussed in the context of the structures IIa vs. IIb shown in Table 2. An extended discussion on such aspects and potential implications for larger particle sizes (tens to hundreds of nm diameter) would be of interest to the community.

We also expected the formation of a single region, but in our simulations for this system we got twice the same three regions. In both simulated systems, the hydrophilic regions were quite far away so as not to interact due to the intermediate hydrophobic region. Starting from the moment of the particle formation the system's potential energy decreased gradually and reached a plateau at the middle of the simulation period as the particle was approaching the final configuration. The system's energy fluctuated (plus or minus 2 percent) around this low energy value for the second half of the simulation period. The difference of the potential energy between the two final configurations in the repeated simulations of this system was less than 1 percent. Of course, given the duration of our simulations we cannot guarantee that the three aqueous structures will not merge at much longer timescales. Please note that while the presence of these three regions even at short timescales is intriguing, the main characteristic of these particles is not the exact number of the hydrophilic regions, but that fact that the particles are inhomogeneous and phase separated. We did simulate a few systems (e.g., systems 14 and 25, the particle of simulation 25 is shown in Figure 11d) twice as long and we did not observe any changes in the particle morphology. System 14 is one of the structures of type II mentioned by the referee. The above discussion has been added to the revised manuscript.

**Specific comments**

**(6)** Page 2, lines 20 - 22: Consideration of the effect of organic coatings/films/phases on the formation of cloud droplets have also been central topics in key studies by Ruehl et al. (2014; 2016) and Ovadnevaite et al. (2017).
We have added the Ruehl et al. (2014) and Ovadnevaite et al. (2017) references and corresponding discussion to the Introduction.

**(7)** Page 3, line 1: "surface partitioning" is bulk–surface partitioning meant?
Yes. It is clarified now in the revised text.

**(8)** Page 3, lines 12 - 19: In addition to the cited experimental studies, the application of thermodynamic equilibrium models have also provided valuable insights into the factors of organic–inorganic interactions and composition ranges of liquid–liquid phase separation, e.g. the works by Zuend et al. (2010), Zuend and Seinfeld (2012), Renbaum-Wolff et al. (2016), Pye et al. (2018). Conclusions from such studies for the mixing in (non-nanoscale) aerosols are in agreement with the statement made in the

last paragraph of Section 4 in this article. The authors may want to extend the discussion on Pages 3 and 21 considering those studies.
This is a good point. A discussion of the above publications has been added in the Introduction but also in the Conclusions.

**(9)** Page 4, line 2: "theoretical and kinetic models". What do you mean by theoretical; are you referring to thermodynamic models? The latter are not more or less "theoretical" than kinetic models.
We have rephrased this rather confusing sentence.

**(10)** Page 4, line 29: phrasing: "liquid either glassy"
We have followed the reviewer's suggestion.

**(11)** Page 5, line 21: spelling: carboxyl (not carboxylic)
The typo has been corrected.

**(12)** Page 16, line 12: I think "Fig. 11d-e" should be Fig. 10d-e.
The figure number was corrected.

**(13)** Page 17, lines 1-4: Fig. 11 should be Fig. 10 in all instances there.
The figure numbers were corrected.

**(14)** Page 20, line 9: correct phrasing of "salt aqueous solutions".
We have rephrased this.

**(15)** Page 20, line 15: in this comparison with experimental studies, especially regarding a potential size effect on phase separation, it would be useful to further discuss the simulation results compared to the experimental findings under different particle drying rates by Veghte et al. (2013) and by Altaf and Freedman (2017).
We have followed the reviewer's suggestion and added the corresponding discussion.

**(16)** Fig. S23 of the Supplement is neither discussed/mentioned in the main text nor the supplement text.
Figure S23 is now mentioned in Section 3.5.2.

**Responses to the Comments of Reviewer 2**

**(1)** This manuscript describes the morphology of mixed organic-inorganic nanoparticles under different composition regimes and environmental conditions (particularly RH) according to molecular dynamics simulations. The methods draw heavily upon a previously peer-reviewed paper by the first author (from 2017). The manuscript is written clearly and presents a concise summary of the diverse morphologies that can result from their prescribed conditions. The material is relevant for the ACP community and merits publication after the following comments have been addressed, many of which will likely not affect the conclusions.

We thank the reviewer for his/her comments. Our responses (in regular font) and corresponding changes to the manuscript are given after each comment (in italics).

**(2)** For a volume that corresponds to 1 atm and 320 K for the system sizes studied, should there not be about 3 to 9 water vapor molecules per 100 M molecules (N2 or O2) for the RHs described?

The estimate of the reviewer is correct. Based on this we reexamined the partitioning of the water molecules between the gas and particulate phases focusing on the water molecules that were found near the particle surface. These molecules were included in the particulate water initially. We have corrected this in Table 1 and changed the average number of water molecules in the two phases accordingly. A brief discussion of this point has also been added to the revised paper.

**(3)** Is a vapor phase necessary at all? While NPT at 1 atm better reflect experimental and ambient conditions, the barostat used provides isotropic dimension rescaling (i.e., volume rescaling). When simulating a condensed-phase surrounded by "gas", this rescaling should have little effect on the condensed phase as the gas-gas and gas surface interactions are infrequent over the simulation timescales used in this work (approximately 1 collision per 10 ns?). The condensed-phase properties are therefore maintained primarily by the force field parameterizations, and similar results should be obtained with NVT simulations with no gas-phase molecules for these "low" pressures (and is more often used in such studies).

We agree with the reviewer that for the reasons mentioned the effect of the gas phase in these simulations is probably of secondary importance. However, it is not negligible. The frequency of collisions for example for system 1 was 2 collisions per 10 picoseconds (much higher than the estimated value above). Apart from the force field choice, the statistical ensemble choice is also important for a MD simulation. The *NpT* statistical ensemble used allows more precise condensed–phase density calculations. The *NVT* statistical ensemble is usually chosen when the simulation focuses on conformational properties in vacuum without periodic boundary conditions. The periodic boundary conditions are essential for the electrostatic energy calculations in our simulations. The above discussion has been added to the paper.

**(4)** Is acid deprotonation neglected in these simulations?

The organic acids used in the investigated systems are quite weak, so their deprotonation has been neglected. This is now explained in the manuscript.

**(5)** While SPC/E is commonly used, other researchers have previously found polarizable force fields for water to be important for determining distributions of sodium and chloride ions in a system with an air/water interface (Jungwirth and Tobias, 2000). Some mention should probably be made especially for interpreting the morphology from the organic-inorganic-water system?

SPC/E is a simple model with low computational cost, which includes a term for the polarization energy. We believe that a polarizable force field for water would not result in different morphologies in our cases. The distributions of sulfate and ammonium ions inside the simulated particles are in agreement with those reported by Jungwirth and collaborators (Gopalakrishnan et al., 2005; Jungwirth et al., 2005), who have used polarizable models for ammonium sulfate aqueous solutions. The repulsion of all the ions from the surface region, the ammonium slight preference for the surface, and the increased concentrations of ions near the core reported in the corresponding studies are also observed in our study. Following the reviewer's suggestion, we have added a brief discussion of this topic.

Gopalakrishnan, S., Jungwirth, P., Tobias, D. J. and Allen, H. C.: Air-liquid interfaces of aqueous solutions containing ammonium and sulfate: Spectroscopic and molecular dynamics studies, J. Phys. Chem. B, 109, 8861–8872, 2005.
Jungwirth, P., Rosenfeld, D. and Buch, V.: A possible new molecular mechanism of thundercloud electrification, Atmos. Res., 76, 190–205, 2005.

**(6)** While difficult to generalize these findings at this stage, one conclusion put forth is that the diversity in heterogeneities can result but are currently not well-represented by chemical transport models. On this point, sometimes it is unclear what are surprising findings from this work vs. what was known previously (from experiment or MD simulation). The phase separation of organics by O:C is discussed by the authors, but the dependence of bulk-surface partitioning of mildly soluble species on concentration and composition is of course known (e.g., also demonstrated in molecular simulation by Hede et al. 2011 cited by the authors), and is anticipated by the Gibbs adsorption isotherm (and more recently by a finite volume model - Malila and Prisle 2018). Phase separation among immiscible organics have been investigated by Ye et al. (2016). Organic islands have been found in experiments (Garland et al. 2008) and in simulation (Hede et al. 2011). Also, aspherical droplets due to water-organic interactions have previously been reported in several papers by Zachariah and co-workers (including that already cited by the authors in the introduction).?

Our comment here was related to chemical transport models that assume that all or at least all the secondary organics in the particles dissolve in the aqueous phase. The reviewer is correct that there has been a lot of work in previous experimental and theoretical studies, but this work has not found its way yet to most chemical transport models. This is now explained better in the revised paper. While parts of our work are in agreement with existing experimental results (this is encouraging) and other theoretical studies using rather different approaches, this is one of the first efforts trying to examine relative realistic atmospheric particle compositions and multiple organic compounds. This approach promises to synthesize the previous studies which had more limited scope. The most surprising result of our work is probably the

structure of the particles containing multiple organics. The discussion in Section 4 was extended so as to include the above points.

**Insights into the morphology of multicomponent organic/inorganic aerosols from molecular dynamics simulations**

Katerina S. Karadima,[1,2] Vlasis G. Mavrantzas,[1,2,3] Spyros N. Pandis[1,2,4]

[1]Department of Chemical Engineering, University of Patras, Patras, GR 26504, Greece

[2]Institute of Chemical Engineering Sciences (ICE–HT/FORTH), Patras, GR 26504, Greece

[3]Department of Mechanical and Process Engineering, ETH Zürich, CH 8092 Zürich, Switzerland

[4]Department of Chemical Engineering, Carnegie Mellon University, Pittsburgh, PA 15213, USA

*Correspondence to*: Spyros N. Pandis (spyros@chemeng.upatras.gr)

**Abstract.** We explore the morphologies of multicomponent nanoparticles through atomistic molecular dynamics simulations under atmospherically relevant conditions. The particles investigated consist of both organic (cis-pinonic acid/CPA, 3-methyl-1,2,3-butanetricarboxylic acid/MBTCA, $n$-$C_{20}H_{42}$, $n$-$C_{24}H_{50}$, $n$-$C_{30}H_{62}$ or mixtures thereof) and inorganic (sulfate, ammonium and water) compounds. The effect of relative humidity, organic mass content and type of organic compound present in the nanoparticle is investigated. Phase separation is predicted for almost all simulated nanoparticles either between organics and inorganics or between hydrophobic and hydrophilic constituents. For oxygenated organics, our simulations predict an enrichment of the nanoparticle surface in organics, often in the form of islands depending on the level of humidity and organic mass fraction, giving rise to core–shell structures. In several cases the organics separate from the inorganics, especially from the ions. For particles containing water–insoluble linear alkanes, separate hydrophobic and hydrophilic domains are predicted to develop. The surface partitioning of organics is enhanced as the humidity increases. The presence of organics in the interior of the nanoparticle increases as their overall mass fraction in the nanoparticle increases, but this depends also on the humidity conditions. Apart from the organics–inorganics and hydrophobics–hydrophilics separation, our

simulations predict a third type of separation (layering) between CPA and MBTCA molecules under certain conditions.

**1 Introduction**

Atmospheric aerosol particles exert a major impact on human health (World Health Organization, 2016), air quality (Fuzzi et al., 2015), and visibility (Wang et al., 2015). They also impact Earth's radiation balance and consequently global climate, but their contribution to climate forcing remains highly uncertain (IPCC, 2013).

Atmospheric particles typically contain both organic and inorganic compounds. The fine inorganic mass fraction is mainly inorganic salts, with ammonium sulfate being the most abundant (Zhang et al., 2007). The contribution of organics to sub–micrometre atmospheric particulate matter mass can be as high as 90%. The thousands of atmospherically–relevant organic compounds differ significantly in volatility, solubility and other physicochemical properties, and their complexity hinders our overall understanding of atmospheric organic aerosol (Glasius and Goldstein, 2016).

During its lifetime, atmospheric particulate matter evaporates or grows by condensation but also undergoes chemical transformations due to a number of heterogeneous reactions (Buseck and Adachi, 2008; Seinfeld and Pandis, 2006). Aerosol phase and morphology, among others, can play a significant role in these processes. Specific morphologies, such as coating by organic films, may affect water uptake (Davies et al., 2013) and, the rate of heterogeneous reactions (Folkers et al., 2003; Zhou et al., 2012). Recent studies (Sareen et al., 2013; Ruehl and Wilson, 2014)or the formation of cloud droplets (Ovadnevaite et al., 2017; Ruehl and Wilson, 2014; Sareen et al., 2013).  haveThe last studies presented evidence that athe hydrophobic organic–rich surface not only results in lowerthe surface tension for the particlelowering and, but also increases the particledroplet hygroscopicity (Ruehl and Wilson, 2014; Sareen et al., 2013), 
[revised manuscript text omitted]

~~The RH was furthermore estimated by taking into consideration the water molecules in the gas phase (Table 1). For this calculation the partitioning of the water molecules between the gas and particulate phases should be examined carefully so as not to include the water molecules that are found near the particle surface. In all cases the RH is just a few percent different than the value based on particle thermodynamics. However, the RH calculation based on the number of water molecules is quite uncertain because there are only a few such molecules in the simulation cell (e.g. in simulation 25, if there were 4 water molecules in the gas phase the resulted RH would be 46 %).~~

~~At the particle diameter scale examined here and due to the high curved liquid vapour interface, one would expect that the Kelvin effect is non negligible. However, the estimation of the magnitude of the Kelvin effect in these particles is quite uncertain given that in a lot of the investigated systems (e.g., those that include cis-pinonic acid) the organic acts as a surfactant. As a result, the magnitude of the Kelvin effect, depending on the assumed surface tension, can be up to 60 % higher that the bulk equilibrium RH values (Table 1), but also much lower, just up to a few percent. On the other hand, the estimated RH values based on the concentration of the~~

[revised manuscript text omitted]

[1]All RH values are rounded to the nearest integer.

[revised manuscript text omitted]